# Modified Polysaccharides: Potential Biomaterials for Bioprinting

**DOI:** 10.3390/jfb16090338

**Published:** 2025-09-09

**Authors:** Tao Jiang, Yun Yang, Zening Lin, Yang Hong, Zirong Luo

**Affiliations:** 1College of Intelligence Science and Technology, National University of Defense Technology, Changsha 410073, China; yangyun17@nudt.edu.cn (Y.Y.); linzening@nudt.edu.cn (Z.L.); hongyang16@nudt.edu.cn (Y.H.); 2National Key Laboratory of Equipment State Sensing and Smart Support, National University of Defense Technology, Changsha 410073, China

**Keywords:** polysaccharides, modification methods, 3D bioprinting, printability, tissue engineering

## Abstract

Polysaccharides have emerged as promising biomaterials for 3D bioprinting due to their inherent biocompatibility, biodegradability, and structural diversity. However, their limited mechanical strength, insufficient bioactivity, and suboptimal printability hinder their direct application in fabricating complex tissue constructs. This review systematically summarizes universal modification strategies to address these challenges by tailoring polysaccharides’ physicochemical and biological properties. We first analyse the fundamental requirements of bioprinting materials, emphasising on the critical role of shear-thinning behaviours, post-printing structural fidelity, and cell-instructive functions. Subsequently, we highlight the advantages and limitations of representative polysaccharides, including chitosan, alginate, and hyaluronic acid. Chemical functionalisation, physical reinforcement, and biological hybridisation are proposed as versatile approaches to synergistically enhance printability, mechanical robustness, and bioactivity to tackle the limitations. Furthermore, dynamic crosslinking mechanisms enabling self-healing and stimuli-responsive behaviours are discussed as emerging solutions for constructing biomimetic architectures. Finally, we outline future directions in balancing material processability with cellular viability and scaling up modified polysaccharides for clinical translation. This review aims to provide a design blueprint for engineering polysaccharide-based bioinks toward next-generation regenerative medicine.

## 1. Introduction

In recent years, three-dimensional (3D) bioprinting [1,2,3,4] has attracted wide research interest in tissue engineering and clinical applications. This technology allows for unmatched architecture control, adaptability, and repeatability that can overcome the limits of conventional biofabrication techniques. We take extrusion-based 3D bioprinting as an example, the basic principle of which is to stack bioinks layer by layer following programmed paths to form 3D bioactive tissue with designed geometries [5], where the bioinks are biomaterials encapsulated with living cells. Bioprinting can quickly fabricate large-scale structures with high resolution, giving it great potential for tissue engineering [6,7,8,9,10]. The mechanical and biological properties of bioinks are key elements in bioprinting that determine structural stability and cell survivability and have triggered great interest in the field to seek the so-called “ideal” bioinks. To ensure the survival of cells, bioinks need to be biocompatible. To this end, natural polymer materials (polysaccharide [11,12,13], protein [14,15]) and synthetic polymer materials (polyethylene glycol [16]) are developed. Natural polymer materials that recapitulate the extracellular matrix (ECM) in structure and function are the easiest to recognise and accept by in vivo organisms [17]. The natural polymer materials consist predominantly of polysaccharide compounds (e.g., glycosaminoglycans, proteoglycans, and glycoproteins) [18], which support cellular growth and promote tissue regeneration. The polysaccharide-based functional materials thus have great application potential in tissue engineering and have been widely studied in the past decades.

Polysaccharides are one of the most common substances in nature, and play an extremely key role in organisms [19], including energy storage and structural support [20]. Benefiting from their availability and handleability, polysaccharides are considered a high-value renewable material [21]. The basic chemical structure of polysaccharides is relatively simple: monosaccharide molecules are sequentially linked by O-glycosidic bonds formed by dehydration and condensation. The source of materials and the degree of polymerisation influence the chemical structure, functional groups, molecular weights, and electrical charges of the resultant polysaccharides, leading to a great diversity of polysaccharides with completely different mechanical and biological properties. Despite the many outstanding properties of natural polysaccharides, such as biocompatibility, biodegradability, and hydrophilicity, most natural polysaccharides inevitably have drawbacks in particular aspects. For example, chitosan has poor solubility [22], alginate shows weak mechanical stability [23], and agarose exhibits an excessive gelling temperature for cells [24], which all hinder them from being directly used in bioprinting. On the other hand, the large numbers of chemically active functional groups [25] such as -COOH, -NH_2,_ and-OH found in natural polysaccharides have enabled them to be physically or chemically modified on demand to achieve significantly improved mechanical and biological properties [26], making them candidate bioinks for bioprinting.

Herein, we systematically review conventional methodologies employed for the modification of diverse polysaccharides to enhance their suitability for bioprinting applications, while concurrently establishing comprehensive evaluation frameworks for assessing their bioprintability (Figure 1). Polysaccharides are categorized according to their biological origins into three primary classes: plant-derived, animal-derived, and microbial-derived polysaccharides. Initially, we elucidate the fundamental structural characteristics and physicochemical properties of these polysaccharides, followed by an in-depth analysis of potential modification strategies to optimise their performance in bioprinting and tissue engineering applications. Subsequently, we delineate the fundamental principles and critical parameters governing 3D bioprinting processes specifically pertaining to polysaccharide-based bioinks, with particular emphasis on quantitative printability assessment methodologies. Furthermore, we critically examine representative applications of modified polysaccharides in tissue engineering through bioprinting technologies. We anticipate that this comprehensive analysis will not only provide valuable insights for future advancements in polysaccharide research and derivative development, but also facilitate the expansion of these versatile biomaterials into applications extending beyond the realm of tissue engineering.

## 2. Classification and Modification of Polysaccharides

Polysaccharides have emerged as versatile biomaterials with broad applications across multiple disciplines, owing to their structural diversity, functional versatility, and abundant natural sources. Their inherent biomimicry of extracellular matrix (ECM) components renders them particularly attractive for tissue engineering applications. While native polysaccharides exhibit several advantageous characteristics—including excellent biocompatibility, pronounced hydrophilicity, and negligible toxicity—they are frequently limited by intrinsic drawbacks such as inadequate mechanical stability, suboptimal printability, and absence of bioactive functional groups.

To address these limitations, the diverse and tunable structures of polysaccharides enable strategic modifications to tailor their properties for specific applications. In this section, polysaccharides are systematically classified into three categories based on their biological origin: plant-derived, animal-derived, and microbial-derived polysaccharides. For each category, we analyse their fundamental physicochemical properties and inherent limitations, while proposing potential modification strategies (Table 1) to enhance their performance in bioprinting and tissue engineering applications.

### 2.1. Plant Polysaccharides

#### 2.1.1. Cellulose

Cellulose [52,53] (Figure 2a) is the world’s most abundant renewable natural polymer and exists in many plant species. Many studies have indicated excellent advantages in terms of low cytotoxicity, low density, high hardness, and biocompatibility [54]. Since its discovery, it has been considered a polysaccharide with broad prospects in biomedicine, particularly as a bioink. It is a linear semicrystalline polysaccharide composed of D-glucose units connected by β (1–4). Hydrogen bonds and van der Waals forces are formed between cellulose chains when polymers are formed. Therefore, they exhibit desirable features such as high strength, rigidity, thermal stability, and low expansion coefficients [55]. However, the inherent limitations of cellulose as a bioink significantly constrain its biomedical applications, including poor aqueous solubility, incompatibility with hydrophobic polymers, inadequate dimensional stability under physiological conditions, and excessive hygroscopicity. Additionally, native cellulose exhibits restricted biodegradability in mammalian systems, being primarily degradable only by specific symbiotic bacteria and microbial cellulases. To overcome these challenges, strategic modifications are essential to (1) enhance its mechanical robustness through physical crosslinking or composite formation, (2) introduce bioactive functional groups to improve cellular interactions, and (3) optimise its physicochemical properties to ensure optimal biocompatibility and support cell viability. These modifications are crucial for transforming cellulose into a clinically relevant biomaterial for bioprinting applications.

Nanocellulose particles are produced through acid hydrolysis, which breaks hydrogen bonds and amorphous fibrous regions, resulting in nanoparticles (NPs) with well-defined shapes that exhibit high viscosity and shear dilution during printing. Most importantly, the mechanical properties of bioinks can be adjusted [56]. Torres-Rendon [57] printed rod-like structures using cellulose nanocrystals through bioprinting. Highly efficient and compatible cellulase degrades rod-like structures into hollow tubular structures, where mouse fibroblasts can grow and form a fused cell layer.

Cellulose nanofiber (CNF) is another cellulose derivative featuring enhanced mechanical properties, which attracts significant interest as a multifunctional biomaterial. CNF, as a biocompatible additive, is mixed with gelatin–alginate bioinks to improve the printability [58]. Results show that the rheological properties and printability of the bioink are greatly enhanced with the addition of CNF. CNF can also prepare bioinks compatible with pneumatic-based direct ink writing (DIW) [59].

Cellulose possesses abundant modifiable functional groups—particularly hydroxyl and carboxyl moieties [60]—that enable diverse chemical derivatisation strategies, including esterification, oxidation, polymer grafting, and etherification. Among these modifications, carboxymethyl cellulose (CMC) has emerged as a particularly valuable derivative, serving as both a viscosity modulator for bioinks [28] and a versatile platform for composite development.

The incorporation of CMC with complementary biopolymers (e.g., alginate, gelatin, chitosan) yields hybrid bioinks with optimised rheological properties, significantly enhancing printability through synergistic interactions. Furthermore, covalent crosslinking of CMC networks can substantially improve the structural integrity of bioprinted constructs. Notably, CMC-collagen peptide composites demonstrate dual functionality: (1) enhanced cellular adhesion through integrin-binding motifs and (2) improved oxidative stability [60]. Three-dimensional scaffolds fabricated from these composites effectively promote cell adsorption and spheroid aggregation [61,62], while exhibiting exceptional biocompatibility and tunable mechanical properties—critical attributes for advanced tissue engineering applications.

#### 2.1.2. Alginate

Alginate [63,64] (Figure 2b) is a natural anionic biopolymer found mainly in brown algae’s cell walls and intercellular mucilage. It is an unbranched polysaccharide linked by a 1–4 glycosidic bond consisting of β-D-mannuronic acid (M) and α-L-gulonic acid (G) [65]. The G and M sequences of alginate from different sources differ greatly [66], which is why alginate’s physical and mechanical properties can vary significantly. The molecular structure of alginate is similar to that of its source. For example, alginate derived from algae has a wide range of relative molecular masses, whereas alginate derived from bacteria exhibits a high degree of polymerisation. Alginate has high viscosity in the presence of divalent cations [67]. Alginate is an ideal biomaterial because of its low toxicity, biodegradability, and biocompatibility. There are many reasons for these properties: (1) monovalent cations can solubilize sodium alginate gels crosslinked using cations; (2) such gels lack bioactive moieties and thus have poor cell-adhesion properties [68]; (3) they have weak mechanical properties when hydrated in water [69]; and (4) they have low electrical conductivity, thermal conductivity, and non-antibacterial properties.

As mentioned above, based on the number and length variation in G and M [70], the mechanical properties and gelation capabilities of alginates can be modified significantly to meet the requirements of bioprinting. When G-blocks crosslink with Ca^2+^ via the “egg-box” model, a high G/M ratio (>1.5) forms a rigid gel network, enhancing compressive strength. Conversely, M/G > 1.5 creates a looser network, increasing elongation (>50%) but reducing stiffness [71,72].

Alginate bioink made with high and low molecular weights in specific proportions has faultless processing performance [73]. Concentration is another factor, and most studies indicate that the 2–4% range is the optimal window in terms of printability and shape fidelity.

Mixing with other polymers is a vital alginate modification method [74,75]. Seunghyun prepared a composite bioink with alginate, tempo-oxidised cellulose nanofibrils (TOCNFs), and polydopamine NPs (PDANPs), which combined the advantages of each component to achieve superior printability, mechanical properties, and osteogenic properties [76]. The most suitable formulations for 3D cell printing and osteogenic differentiation were 1.5% alginate, 1.5% TOCNFs, and 0.5% PDANP. Additionally, printability was significantly improved through the hybridisation of alginate with carboxymethyl cellulose and montmorillonite clay [77]. To ensure printability and shape fidelity, rheological and swelling tests, filament breakage tests, and melting tests were performed to optimise the material composition. As the most common modifier for alginate, gelatin has been extensively used in cell-laden extrusion-based bioprinters with the advantage of biocompatibility, bioactivity, and its reversible gelation kinetics, supported by both in vivo and in vitro experiments [61,62]. Bin Yao [78] incorporated alginate lyase into alginate–gelatin bioinks to improve their degradation properties and acquire lower stiffness and higher porosity.

The RGD (arginine–glycine–aspartate) tripeptide sequence, naturally present in extracellular matrix proteins like fibronectin and laminin, serves as an effective modifier to enhance the bioactivity of alginate biomaterials. Research by Yuanjia Zhu et al. [79] demonstrated that RGD-modified sodium alginate, particularly at higher concentrations, significantly improves cellular responses through three key mechanisms: enhanced cell viability via strengthened integrin-mediated cell–matrix interactions, promoted integrin clustering that facilitates focal adhesion formation, and accelerated proliferation rates through activation of downstream signalling pathways.

#### 2.1.3. Agarose

Agarose [80,81] (Figure 2c) is a natural linear polysaccharide composed of repetitive units of D-β-galactose (D-Gal) and 3,6-anhydro-α-L-galactose (L-AHG) [82]. Natural agarose (NA) hydrogel demonstrates excellent biocompatibility and unique physicochemical properties, including an intrinsic porous network structure, remarkable plasticity, and thermosensitive gelation behaviour. These characteristics render it particularly suitable for various biomedical applications, such as injectable hydrogels, self-healing biomaterials, and 3D-printed tissue scaffolds. The gelation process exhibits distinct thermal hysteresis, with the sol–gel transition occurring at a lower temperature than the gel–sol transition. This endows agarose with significant advantages as a physical hydrogel that maintains long-term stability at physiological temperatures without requiring chemical crosslinking agents. However, the relatively high melting temperature (>70 °C) necessary for dissolution poses challenges for cell-laden applications, as it may compromise cellular viability. Furthermore, while agarose demonstrates a high water absorption capacity, its inherent biological inertness and mechanical stiffness significantly limit cellular adhesion, proliferation, and differentiation. These limitations underscore the need for strategic modification of agarose to enhance its bioactivity while preserving its favourable mass transport properties for oxygen and nutrient diffusion.

Modification for agarose mainly includes physical and chemical methods. Physical methods modify physical properties such as concentration, and chemical methods alter agarose’s structure and composition (e.g., affixing functional groups). The combination of collagen, chitosan, bacterial cellulose, and agarose can effectively mimic an extracellular matrix. Qiang Zou [83] used three polysaccharide hydrogel composites of nanocellulose, agarose, and sodium alginate with seeded cells as bioinks and polyvinyl alcohol (PVA) as sacrificial material to construct complex structures. Analogously, Eman Mirdamadi [84] prepared agarose–alginate bioinks to print a high shape fidelity structure comparable to Pluronic without additional crosslinkers or sacrificial materials. Interestingly, this bioink could be used for extrusion-based 3D bioprinting, which may provide researchers with an easy-to-manufacture technique for developing complex engineered cartilaginous tissues.

Using native agarose, TEMPO, NaBr, and NaOCl as raw materials, carboxymethylated agarose (CA) was synthesised by chemical methods in Aurelien Forget’s laboratory [85]. CA hydrogels exhibited viscoelasticity and shear-thinning properties over a range of shear rates from 0.1 to 100 rad/s, presumably owing to the organization of CA into a β-sheet-like structure and low intermolecular H-bond formation. This rheological property allows CA to be extruded at room temperature. Yixue Su’s work shows that Pluronic F-127 was used as the fugitive bioinks, and CA as the support bioinks were used to print vessel channels [86].

### 2.2. Animal Polysaccharides

#### 2.2.1. Hyaluronic Acid

Hyaluronic acid [87,88] (Figure 2d) is a negatively charged multifunctional polysaccharide composed of alternating disaccharide units of D-glucuronic acid and N-acetyl-D-glucosamine. It is an unbranched and unmodified polymer with a high molecular weight, determining its biological properties [89]. Natural hyaluronic acid exhibits excellent biocompatibility and bio-adhesion. However, the high degradation rate in living tissues and poor mechanical stability are indisputable. As a biomaterial, it has two important characteristics: (1) a passive molecular structure [90] and (2) functional groups with chemical modifications. It is unlikely that unmodified hyaluronic acid is suitable for bioink due to the lack of shape retention capability. There are many chemical modification methods available, including crosslinking and grafting. However, more researchers have directed their attention toward physical blending. For example, it is common to mix polysaccharides with positively charged proteins or surfactants to enhance their integrity [91].

The performance of hyaluronic acid (HA)-based bioinks in 3D bioprinting is critically dependent on three key parameters: adhesion properties, mechanical integrity, and controlled degradability, all of which significantly influence cellular viability and tissue development. The coordinated regulation of cell adhesion, mechanical integrity, and degradability constitutes a fundamental paradigm in tissue engineering design, collectively governing cellular survival, proliferation, and tissue regeneration. Mechanistically, (1) cell-adhesion-mediated integrin signalling suppresses apoptosis while activating PI3K-Akt pathways to promote cell cycle progression, with RGD peptides and other ligands directing stem cell differentiation; (2) mechanical integrity transduces biophysical cues via mechanotransduction pathways, where substrate stiffness dictates cellular responses, and structural collapse-induced stress shielding triggers apoptosis; (3) precise spatiotemporal control of degradability is critical—enzymatically cleavable matrices enable responsive growth factor release for vascularization, whereas hydrolytic degradation byproducts may elicit inflammatory responses, with excessive degradation compromising structural integrity and insufficient degradation impeding cellular infiltration while promoting fibrotic encapsulation. The strategic integration of these tripartite factors enables dynamic orchestration of pro-survival signalling, lineage specification cues, and tissue remodelling processes, thereby engineering an optimised regenerative microenvironment. Achieving optimal printability requires precise optimisation [92] of hyaluronic acid (HA) concentration (typically 2–10% *w*/*v*) and molecular weight (76–1550 kDa), as these parameters directly govern hydrogel formation kinetics and network density [93]. These optimal ranges may vary significantly, depending on specific bioprinting applications and desired scaffold properties.

To further enhance bioink performance, various modification strategies have been developed, including polymer blending with compounds such as methoxy polyethylene glycol (mPEG) and nanoparticles for precise tuning of rheological behaviour and mechanical properties, as well as gamma irradiation treatment, which provides a sterile and efficient crosslinking method for property modulation. These approaches collectively enable the development of tailored HA-based bioinks that meet the stringent requirements of specific tissue engineering applications while maintaining favourable biological responses.

Chemical modification is also popular in addition to physical modification. From the view of the molecular structure of hyaluronic acid, each disaccharide unit contains four hydroxyl groups, one amide, and one carboxyl group. In principle, two different modification methods can be used: crosslinking [94] and coupling [95]. For the former, hyaluronic acid chains are connected by two or more bonds, so mechanical, swelling, and rheological properties can be altered. Only one bond and special functional units are incorporated to implement labelling for the latter. Both methods can exist simultaneously [96]. Functional group modification is one of the most effective methods. For example, hyaluronic acid modified by mono-alkyl reagents and bis-alkylating agents can be used to construct scaffolds, and esterified hyaluronic acid can be used as a substrate material for the construction of novel supramolecular hydrogel networks [97].

#### 2.2.2. Chitosan

Chitosan [98,99] (Figure 2e) is a natural alkaline polysaccharide produced by the deacetylation of chitin, which contains glucosamine and N-acetylglucosamine in a repetitive unit structure [100]. There are six hydroxyl groups and one amino group in this structure. Chitosan is widely distributed in the cell walls of crustaceans and certain plants. The glucosamine structure and the degree of deacetylation are both important features [101]. The solubility of chitosan increases with the degree of deacetylation. Based on its biocompatibility, biodegradability, and significant antibacterial activity [102], chitosan has numerous applications in the biomedical field, particularly for wound dressing [103]. Chitosan has osteoconductive properties that support the attachments of osteoblasts and the formation of mineralized bones [104]. However, the single-chemical structure of chitosan-based hydrogels [105], weak mechanical integrity, and low structural stability significantly limit their application in 3D bioprinting. Controlling the degree of deacetylation and molecular weight can significantly impact physicochemical properties such as crystallinity, solubility, and degradability. Additionally, there are many reactive functional groups in the molecular chains of chitosan. As a result, physical or chemical modifications have been implemented to obtain chitosan derivatives that meet tissue engineering needs [106].

Various products such as sponges, NPs, and gel particles can be obtained through physical modification such as ultrasonic treatment, ionizing radiation, and mechanical grinding [107]. The cavity effect of ultrasound disrupts the intermolecular or intramolecular hydrogen bonds of chitosan polymers. Based on this principle, Klaypradit [108] used an ultrasonic atomiser to obtain chitosan microencapsulated fish oil, and Choo [109] used an ultrasound-assisted drip irrigation system to prepare chitosan/hemispherical microbeads. Liu [110] combined high-pressure homogenization with wet grinding to break the bond connections in the crystals of chitosan and crushed chitosan particles into nanofibers. To produce mechanically controllable composites, Brysch [111] employed thermomechanical processing to enhance chitosan’s mechanical integrity, demonstrating that while room-temperature pressing fails to achieve consolidation, sintering at 180 °C yields substantial hardening (15 ± 0.7 μHV). This effect is further amplified to 26.1 ± 0.1 μHV at 220 °C with carbon nanostructure (CNS) incorporation, attributable to grain boundary refinement and improved chitosan–CNS interfacial cohesion. Yue [112] combined ultraviolet radiation with ozone treatment to tailor the degradation of chitosan. Ozone/UV irradiation accelerates chitosan degradation through hydroxyl radical generation via ozone photolysis, establishing a temperature-dependent duality in chitosan’s structural stability.

Chitosan also contains amine, acetylamino, and hydroxyl active sites suitable for chemical modification, which significantly improves rheological properties, antibacterial properties, and thermal stability [107]. The most common chemical modification is acylation with organic acid compounds, which breaks the hydrogen bonds in chitosan, changes its initial crystalline form, and increases its solubility [113]. Additionally, carboxymethylation, thioylation, graft copolymerisation, and quaternary ammonium salt modification [114] can improve cell biocompatibility and adhesion. Yang [115] reported a method to combine sulfonic acid groups with the amino groups of chitosan to form aminosulfonate compounds. Additionally, chitosan can be modified by phosphorylation, mercaptan, and alkylation to improve the antibacterial properties [116].

Nevertheless, conventional physical and chemical modification approaches often involve complex procedures, stringent reaction conditions, and potential environmental hazards [117]. In contrast, enzymatic modification has emerged as an eco-friendly and biocompatible alternative for polysaccharide functionalisation. This strategy was exemplified by Aberg et al. [118], who employed tyrosinase to successfully incorporate bioactive peptides into chitosan matrices, leading to substantially enhanced viscoelastic properties. Similarly, Wang et al. [119] demonstrated the efficacy of laccase-mediated crosslinking in creating innovative ternary conjugates of β-lactoglobulin, chitosan, and ferulic acid, which exhibited remarkable antioxidant capacity. These enzymatic approaches not only circumvent the limitations of traditional modification methods but also preserve the intrinsic biocompatibility of polysaccharides while imparting desired functional properties.

### 2.3. Microbial Polysaccharide (Xanthan Gum)

Xanthan gum [120] (Figure 2f) is a high-molecular-weight microbial polysaccharide that is synthesised through the polymerisation of D-glucose, D-mannose, and D-glucuronic acid residues at a ratio of 2:2:1 [121]. The material exhibits a high degree of pseudoplasticity and shear rate-/time-dependent viscosity due to the tunability of hydrophilic groups in its molecular structure [122]. Based on its two-stage quintuple right helix structure, it can stably exist over a wide temperature and pH range. Its O-acetyl and acetone residues can crosslink with Na^+^, Ca^2+^, Fe^3+^, and other cations to form high-performance hydrogels.

Rheological properties, including viscosity and shear thinning, can significantly affect the material’s bioprintability. Experimental results showed that the static viscosity and shear thinning of a xanthan gum solution are related to its concentration and molecular weight [123]. When the concentration exceeds 2000 mg/L, the viscosity exhibits an exponential relationship with the concentration [124]. Furthermore, with the addition of complementary modifiers such as phosphate, sodium lactate, and ethyl lactate, the dynamic viscosity decreases, but the shear-thinning property remains. The charge of xanthan gum can be neutralized in a salt solution, which changes the rheological properties of the gel. Jackson [125] used β-lactoglobulin (β-lg) to induce electrostatic attraction during network formation to modify xanthan gum. It was observed that the β-lg aggregates along xanthan chains in the form of a crosslinking agent, and the gelation process and gel network can be controlled by the ratio of β-lactoglobulin and xanthan chains, leading to tunable kinetic properties of the hydrogel. It has also been found that xanthan gum can be mixed with carbon nanotubes, graphene, and metal oxides [126]. In a magnetic field, NPs have been added to promote the self-assembly of xanthan gum and chitosan [127], and magnetic response polyelectrolyte composite hydrogels were fabricated through in situ ion complexation. These hydrogels benefit the adhesion and proliferation of NIH3T3 fibroblasts in a magnetic field, showing great potential in skin, cartilage, muscle, and connective tissue engineering applications.

Chemical crosslinking offers superior control over material properties compared to physical methods, particularly in tuning elasticity and mechanical strength [125]. This advantage is exemplified in the work of Soumya et al. [128], who developed alginate dialdehyde-xanthan gum composite hydrogels using ethyl orthosilicate as a chemical crosslinker. The resulting scaffolds demonstrated both enhanced mechanical integrity and improved 3T3 fibroblast adhesion and proliferation. Chemical modification strategies such as esterification and etherification have proven particularly effective for optimising xanthan gum properties, including its adhesive characteristics and swelling behaviour. Quan et al. [121] demonstrated this approach through the successful synthesis of hydrophobic xanthan gum via hexadecyl grafting onto the polysaccharide’s hydroxyl groups. Conversely, selective removal of functional groups can also serve as an effective modification strategy, as shown by Pinto et al. [7], who observed that alkaline deacetylation progressively increased viscosity with rising pH. These chemical modification techniques provide versatile tools for precisely engineering polysaccharide properties to meet specific biomedical requirements.

Ionizing radiation has emerged as an effective tool for polymer modification, offering precise control over network structure formation at remarkably low energy inputs. Hayrabolulu et al. [129] demonstrated this capability through the synthesis of xanthan gum hydrogels with tunable network architectures using γ-radiation doses as low as 1–2 kGy in controlled acetylene (C_2_H_2_) and carbon tetrachloride (CCl_4_) atmospheres. This radiation-induced crosslinking approach enables the fabrication of polysaccharide networks with tailored properties while maintaining the biocompatibility essential for biomedical applications. The method’s efficiency at such low radiation doses highlights its potential as an energy-efficient alternative to conventional chemical crosslinking techniques.

## 3. 3D Bioprinting Using Modified Polysaccharides

Additive manufacturing (AM) has been considered to be a promising technology for fabricating artificial tissue and organs for transplantation [130]. AM utilises layered deposition of materials to form complex 3D geometries that are difficult to manufacture via conventional subtractive methods [131]. Bioprinting, a branch of it, is dedicated to biomedicine and bioengineering applications, and has boosted the development of biofabrication in the past decade. A variety of 3D bioprinting methods using modified polysaccharides have been developed, including stereolithography [132], inkjet [133], laser-assisted [134], and extrusion-based bioprinting [135]. The inherent rheological and photosensitivity limitations of polysaccharides significantly increase their adaptation challenges for non-extrusion bioprinting technologies. Consequently, current modification strategies predominantly focus on extrusion-based bioprinting, which remains the most prevalent method due to its exceptional bioink compatibility (across a wide spectrum of formulations) and multi-material printing capability. This review specifically concentrates on quality assessment metrics for extrusion-based systems, as other bioprinting mechanisms have been extensively covered in the existing literature [136,137,138,139].

### 3.1. Factors Affecting the Printing Performance of Modified Polysaccharides

The factors affecting the printing performance of biological ink stem from two aspects, namely (1) the characteristics of bioink itself [140], particularly rheological properties, and (2) the parameters of the printing processes.

Rheological properties such as shear thinning, viscoelasticity [141], and yield stress [142] in combination depict the deformation and flow behaviours of the material under external stress, which significantly impact the printability of a bioink. In an extrusion printing system, a bioink changes from an amorphous-dominant state before extrusion to a liquid-dominant state during extrusion due to the lowered viscosity, owing to the shear-thinning property, before returning to an amorphous-dominant state after extrusion [143,144].

On the other hand, printing parameters including nozzle diameter [145], extrusion pressure [146,147,148,149], and translational speed [140,141] can also greatly affect the quality of printing. The influences of these parameters are always coupled. For example, a higher extrusion pressure typically requires a higher translational speed to achieve a preferable printing quality [150], while over-speeding can cause filament discontinuity and lead to failed printing. Similarly, a lowered nozzle diameter would require a higher pressure to initiate the extrusion and a lowered speed to compensate for the decreased flux in the nozzle.

### 3.2. Printability Assessment of Modified Polysaccharides

In extrusion-based bioprinting, deposited bioink filaments can undergo plastic deformation and fusion between adjacent filaments, resulting in decreased geometric resolution and unpredictable deviation from designed structures [151]. Hence, several methods have been proposed to assess the printing fidelity, attempting to identify the factors governing the printing process.

Sotorrí [152] defined printability as transporting a bioink to printer nozzles for layer-by-layer extrusion and deposition at a predictable deformation. Gillispie [153] provided a more intuitive definition: the ability of a material to be printed in a manner that results in desirable outcomes for a given application when subjected to a certain set of printing conditions. Additionally, Kang [154] stated that printability refers to the standardisation of printing accuracy and printing processes, including the selection of materials and the configuration of parameters.

Once a bioink is extruded, the diameter D of the cylindrical extrudate is dependent on the outlet flow rate Q and the translational speed of the nozzle V [155] Equation (1). A simple method to evaluate the printing quality is to measure the diameter of a filament at specific positions and compare them to the design sizes at the same locations (Figure 3a) [156,157]. Lin et al. [158] adopted a normalised evaluation approach, where the widths at 10 points along a filament were measured before calculating their mean $W_{m}$ and standard deviation $W_{SD}$. A dimensionless number $R_{N}$ was then defined according to Equation (2), where a smaller $R_{N}$ value represents better printing quality.(1)$$D=\sqrt{\frac{4Q}{\pi V}}$$(2)$$R_{N}=\frac{W_{SD}}{W_{m}}.$$

Naghieh [158] proposed a method to define the “strand printability ($sP_{r}$)”, as shown in Equation (3), where $D_{s}$ is the designed diameter of a strand and $D_{\exp\;}$ is the measured diameter. The “pore printability ($pP_{r}$)” is described by Equation (4) [159], where L is the perimeter and A is the area of a printed lattice structure. An ideal material and printing setup would result in both $sP_{r}$ and $pP_{r}$ close to one.(3)$$sP_{r}=1-\frac{D_{s}-D_{\exp.\;}}{D_{s}}$$(4)$$pP_{r}=\frac{L^{2}}{16A}.$$

Hazur [160] and colleagues focused on the goodness of printed squared structures and used the following equation to quantitatively describe the printability *P*:(5)$$P=\frac{1}{\left[\left|\frac{1}{2}\times (\frac{{\bar{L}}_{bx}}{L_{ox}}+\frac{{\bar{L}}_{by}}{L_{oy}})-1\right|+1\right]\times \left[1+\frac{1}{2}\times (\frac{SD_{bx}}{L_{ox}}+\frac{SD_{by}}{L_{oy}})\right]},$$
where $L_{ox}$ and $L_{oy}$ are the designed side lengths in *x* and *y* directions, ${\bar{L}}_{bx}$ and ${\bar{L}}_{by}$ are the measured means of lengths, and $SD_{bx}$ and $SD_{by}$ are the standard deviations of the lengths in corresponding directions. *P* can take values between zero and one, while a preferable printability feature is a *P* close to one.

However, current evaluation methods primarily focus on geometric fidelity—the congruence between printed and designed structures—while neglecting critical biological parameters including cell-adhesion efficiency and viability within the bioprinted constructs. We propose that a comprehensive printability assessment framework should incorporate two fundamental dimensions: (1) structural fidelity, encompassing shape accuracy and mechanical stability of the printed architecture; and (2) biological performance, evaluating cellular responses such as adhesion kinetics, proliferation rates, and long-term viability. This dual-aspect evaluation paradigm enables more rigorous and clinically relevant characterisation of bioink printability, bridging the gap between engineering precision and biological functionality essential for successful tissue engineering applications.

In recent years, semi-quantitative evaluations based on pore shape fidelity have gradually emerged, and the dimensionless parameter $P_{r}$ (0 < $P_{r}$ ≤ 1) defined in Equation (6) has become the most common evaluation index (Figure 3b), where L and $A_{a}$ are the perimeter and area of the printed structure, respectively. It is evident that as it approaches 1, the shape fidelity improves significantly [161]. In addition to evaluating the shape fidelity in the x-y plane, it is also necessary to assess the degree of collapse along the Z-axis direction. There is an obvious error between $h_{1}$ (designed) and $h_{2}$ (printed) in Figure 3c. For example, in Michael’s [162] and Ribeiro’s [151] studies, the differences between the designed and printed constructs were clarified (Figure 3d).(6)$$P_{r}=\frac{L^{2}}{16A_{a}}$$

The degree of damage or survival rate of cells after printing is an important index for evaluating the biocompatibility of bioinks. Han [129] conducted numerical and experimental studies on cell damage for pneumatic extrusion-based 3D bioprinting and proposed a predictive function for the damaged cell ratio. The validity of the function was verified via living/dead cell staining, proving its capacity to provide appropriate guidance for 3D bioprinting by selecting proper nozzle geometries and operating pressure.

**Figure 3 jfb-16-00338-f003:**
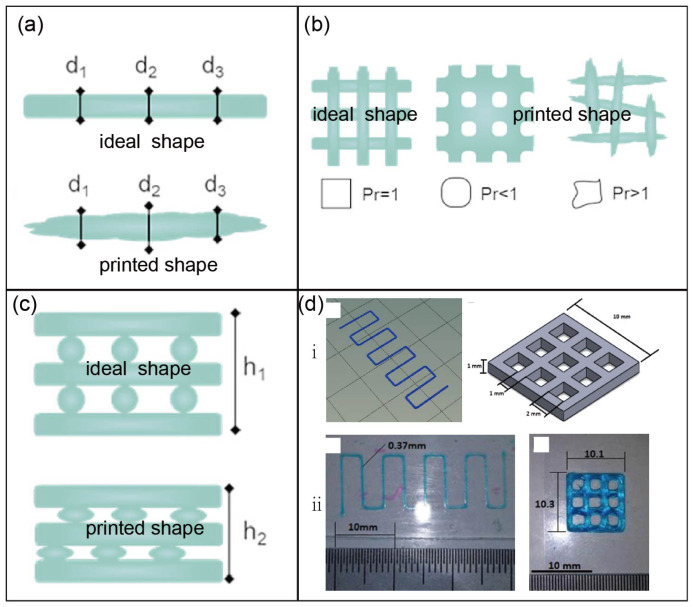
The assessment of shape fidelity: the assessment of cylindrical filament (**a**), pore (**b**), and height along the z-axis direction (**c**) (reprinted from Ref. [143]); (**d**) the differences between the design (**i**) and the printed constructs (**ii**) (reprinted with permission from Ref. [162], copyright 2018, Elsevier).

## 4. Application of Modified Polysaccharides in Tissue Engineering

Polysaccharide-based biomaterials have unique biocompatibility and are highly similar in structure to natural extracellular matrices. The unique structures of polysaccharides make it easy to improve their properties through physical and chemical modification for tissue engineering applications.

### 4.1. Cartilage Tissue Engineering

Chitosan has a significant effect on bone regeneration, yet its insufficient immunogenicity and biodegradability necessitate a modification with growth factors or scaffold materials to render it an effective alternative for autogenous bone transplantation [163]. He et al. [106] prepared a novel chitosan-based hydrogel with ethylenediamine tetraacetic acid and used calcium solution for physical crosslinking. Modified chitosan and pure chitosan were compounded to obtain a bioink with the advantages of rapid gelation and high printing accuracy that can be used to construct 3D chondrocyte scaffolds. Combining nanofibrillated cellulose (NFC) and alginate is suitable for chondrocyte growth. Markstedt et al. [164] successfully fabricated cell-laden human ears and menisci using bioprinting. The viability of human nasoseptal chondrocyte cells significantly dropped from 95.3 ± 0.1% to 69.9 ± 13.3% (*p* < 0.05) after bioink encapsulation but recovered to 72.8 ± 6.0% at day 1 and further increased to 85.7 ± 1.9% by day 7 post-printing. These results demonstrate that their bioink exhibits excellent printability and promotes cartilage regeneration (Figure 4a).

Ear deformities can cause significant psychological issues in patients, and various side effects accompany surgical treatment. Fortunately, biological 3D printing technology can be used to manufacture an auricle. Flégeau et al. [165] first prepared a tyramine-modified hyaluronic acid hydrogel (HA-TYR) and enzymatically crosslinked it by adding horseradish peroxidase and H_2_O_2_. This hydrogel exhibits tunable porosity, shear-thinning, and excellent thixotropic properties, with which an in vitro model of a human ear was bioprinted (Figure 4b).

Meniscal injury is another common disease. Three-dimensional bioprinting minimises the mismatch between artificial meniscus implants and natural knee tissue. Wenbin Luo [58] developed an improved bioink in which cellulose nanofiber (CNF) was mixed with gelatin–alginate bioinks to ensure high-precision bioprinting. They collected MRI information to reconstruct personalised meniscus models, which were then manufactured with CNF-modified gelatin–alginate bioinks (Figure 4c).

**Figure 4 jfb-16-00338-f004:**
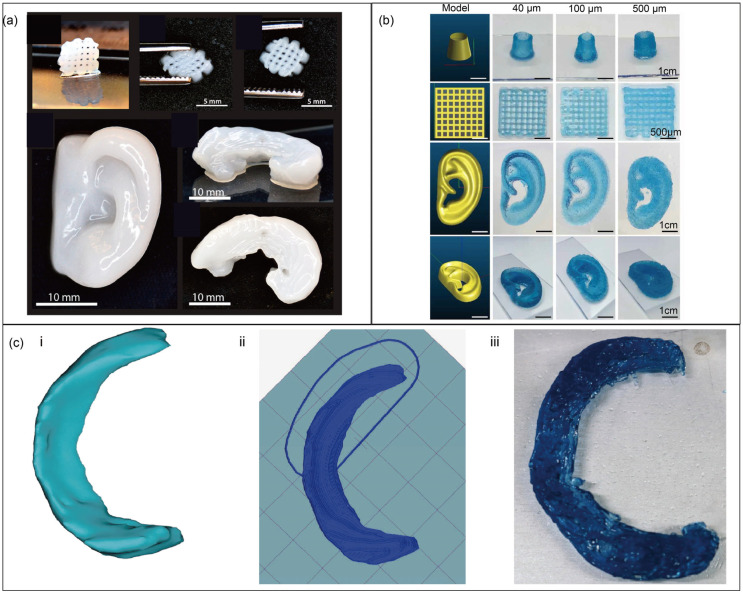
Biofabrication structures: the structures of grid, human ear, meniscus (**a**) (reprinted with permission from Ref. [164], copyright 2015, American Chemical Society), representative objects of 3D-printed objects using HA-TYR bioinks (**b**) (reprinted from Ref. [165]), 3D Meniscus model (**c**) (**i**,**ii**), and bioink-based, cell-free meniscal prototype (**c**) (**iii**) (reprinted from Ref. [58]).

### 4.2. Vascular Tissue Engineering

The vascular system serves as the critical circulatory network responsible for nutrient transport and metabolic homeostasis in living organisms, making its structural and functional integrity essential for survival. Given the substantial global health burden imposed by vascular pathologies—including atherosclerosis, aneurysms, and peripheral artery disease—the development of effective vascular repair strategies represents a pressing clinical imperative.

As early as 2017, Kreimendahl et al. [166] cultured endothelial cells [167] and fibroblasts [168] in a mixture of agarose, type-I collagen, and fibrinogen to form a capillary network. Their work proved that printability and angiogenesis are not mutually exclusive. A year later, Štumberger and Vihar [169] used gelatin and xanthan gum to create complex 3D structures, where xanthan gum was used as a sacrificial material to construct complex channels simulating the vascular structure of tissue (Figure 5a). In 2020, by using Pluronic F127 as a sacrificial component, Li et al. [120] proposed a hybrid biological ink composed of alginate (Alg) and silk fibroin (SF). Through rheological tests, Fourier transform infrared spectroscopy (FTIR) analysis, and cytocompatibility evaluation, this bioink was proven to be a promising coaxially extruded bioink. Based on the sacrifice of Pluronic F127, a hierarchical microchannel network was formed in the structure, laying the foundation for manufacturing large-scale blood vessels.

Antunes et al. [170] prepared an alginate tubular structure (Figure 5b) where the sacrificial component was gelatin. Because there is no natural cell-binding motif in alginate, type-I collagen or silk fibroin was added to improve the bioactivity of alginate-based hydrogels. Additionally, Ba^2+^ was used for secondary crosslinking to ensure stability in an aqueous environment. The mechanical properties were enhanced sufficiently after crosslinking to produce adequate vascular substitutes. Zou [83] used three polysaccharide hydrogel composites of nanocellulose, agarose, and sodium alginate with seeded cells as bioinks and polyvinyl alcohol as a sacrificial material to fabricate structures. The resulting nutrient network gradually formed a blood-vessel-like structure (Figure 5c).

In 2021, Dogan et al. [171] formulated an alginate/collagen type-I bioink supporting extrusion-based bioprinting, where human-induced pluripotent stem-cell-derived mesodermal progenitor cells (hiMPCs) retained their capabilities, giving rise to the formation of complex vessels. Vascular endothelial growth factor was added to encourage the production of blood vessels, and the formation of various vessels within bioprinted scaffolds was observed after 21 days of culture.

### 4.3. Skin Tissue Engineering

Skin, the largest organ of the human body, is the first line of defence against external threats. There are numerous examples of skin trauma that cause pathological changes in the body. Therefore, skin transplantation is an important treatment option. However, when skin is burned in a large area, few transplantable skin sources can meet the resulting demand. Therefore, the development of active skin substitutes with physiological functions has attracted the attention of many researchers. Sandri [172] used chitosan and glucosamine as raw materials and citric acid as a crosslinking agent to create scaffolds for repairing damaged skin. After 18 days of culture, epidermal skin was almost completely rebuilt, exhibiting considerable keratinisation. Su [38] made an agarose–polydopamine (APG) hydrogel with good biocompatibility. Based on the introduction of PDA, the cell migration rate on the surface of the APG was high. Notably, using APG promotes collagen deposition and angiogenesis, which can accelerate the healing of damaged skin.

Alginate has received increasing attention as a raw material for skin production. For example, Somasekharan et al. [173] created a bioink based on alginate blended with gelatin and diethylaminoethyl cellulose (DCEL). The optimal formulation for this bioink is composed of 2% alginate (*w*/*v*), 3.3% gelatin (*w*/*v*), and 0.93% DCEL (*w*/*v*). Fibroblasts and keratinocytes were co-cultured for 21 days, and histological analysis revealed the formation of dermal and epidermal equivalent structures.

### 4.4. Complex Biological Structures

While the complete realization of directly bioprinted functional tissues and organs remains an ambitious objective requiring significant further research and technological advancement, current progress is highly promising. Both commercial entities and academic research institutions are actively engaged in developing increasingly sophisticated in vitro biological constructs, systematically advancing the field toward this transformative medical capability.

As shown in Figure 6a, encapsulated aortic root sinus smooth muscle cells (SMCs) and aortic valve leaflet interstitial cells (VICs) were encapsulated within separate alginate–gelatin hydrogel preparations. These preparations were then used to construct porcine aortic valves using the Fab@Home™ system for bioprinting. After 7 days of culture, SMCs and VICs exhibited over 80% viability [174]. A few years ago, the FRESH technique was developed to create 3D heart structures with complex internal and external architectures (Figure 6b), whose constituent materials included alginate, collagen, and fibrin. Although this construct did not contain cells, it demonstrated the potential of 3D bioprinting and modified polysaccharides for building complex organs [175]. In addition, 3D-printable formulations of hybrid hydrogels, based on methacrylated hyaluronic acid (Me-HA) and methacrylated gelatin (Me-Gel), were developed and used to bioprint heart valve conduits containing encapsulated human aortic valvular interstitial cells (HAVICs) (Figure 6c) [176]. Xuanyi Ma illustrates the first in vitro hepatic model that combinatorially mimics several in vivo features of the liver by providing a 3D culture environment for human-induced pluripotent stem cell (hiPSC)-derived hepatic cells in triculture with supporting cells, arranged in a biomimetic liver lobule pattern [177].

## 5. Summary and Outlook

The development of the ideal bioink formulation is an everlasting challenge. As a common and essential material to make up a creature, polysaccharides have shown remarkable potential in biomedicine, drug delivery, wound dressing, and cancer treatment. However, due to insufficient purity and strength, natural polysaccharides cannot be directly used as biomaterials for bioprinting. To provide a reference for further researchers concerning polysaccharides for bioprinting, this paper reviews the performance characteristics and modification methods for bioprinting, and tissue engineering applications. Chemical modification can modify molecular structures to obtain specific properties, but the operation often requires harsh experimental conditions and produces harmful by-products. Simultaneously, the employment of crosslinking agents and ultraviolet light exposure can harm cellular structures [178]. The operation of physical modification is relatively trivial, with little or no damage to the inherent structure, so it is difficult to change the bioactivity of polysaccharides. Enzymatic modification [179] has peculiarities including a clear objective, consummate manoeuvrability, and outstanding efficiency, but there is a mere handful of available enzymes that limit the application spectrum. In addition, the properties of polysaccharides are highly correlated with their sources and production batches [180]. Although the research on modified polysaccharides for bioprinting has made remarkable achievements, there is undoubtedly still significant room for improvement. Considering these issues, the following perspectives and research directions are provided.

Integrating artificial intelligence (AI) with large models to create an intelligent decision support system for polysaccharide modification is a valuable exploration, with the potential to enhance the efficiency and precision of the modification process. This system first relies on the vast amounts of experimental data already collected, utilising big data techniques for deep integration and cleaning to build a high-precision large model tailored for polysaccharide modification. Subsequently, through comprehensive model training, the system extracts the core features and underlying patterns within the polysaccharide modification process. In practical applications, based on the modification objectives and experimental conditions, the system recommends the optimal modification scheme, including methods and reaction conditions, and accurately predicts the modification outcomes. Furthermore, the system provides real-time monitoring and prediction of critical parameters during the experimental process, ensuring the stability and reproducibility of the entire modification process, thereby significantly enhancing the efficiency and precision of polysaccharide modification. This approach has been proven successful for applications in 3D bioprinting [181,182], biomedical engineering [183], and programmable metamaterials [184].

A critical gap persists in the field, as current methodologies lack robust, quantitative frameworks for objectively evaluating polysaccharide modifications. This challenge demands the establishment of a comprehensive assessment system developed through collaborative efforts among interdisciplinary experts spanning bioprinting, biomaterials science, bioengineering, and clinical medicine. Such a framework should incorporate multidimensional evaluation criteria, including (1) physicochemical characterisation of modified materials, (2) biological performance metrics, and (3) practical processing parameters. Specifically, this system must assess the scientific rationale of modification strategies, technical feasibility of modification processes, and functional printability of resulting materials. Most importantly, rigorous biological validation remains paramount—only through systematic in vitro and in vivo testing can researchers definitively determine whether modifications successfully achieve their intended biomimetic and functional objectives while maintaining essential biocompatibility.

The native physicochemical characteristics of polysaccharides fundamentally govern their modification potential and ultimate performance in 3D bioprinting applications. Critical consideration must be given to three key factors: (1) biological source characteristics (extraction origin and methods), (2) scalability of production processes, and (3) purification efficiency—all of which significantly influence material consistency and functionality. The three primary polysaccharide classifications exhibit distinct structure–property relationships: animal-derived polysaccharides typically offer superior bioactivity but present greater batch variability, plant-based variants provide robust mechanical properties yet require extensive processing, while microbial polysaccharides enable precise molecular control but may necessitate genetic engineering for optimal performance. These inherent advantages and limitations dictate their respective suitability for specific biofabrication applications, ranging from soft tissue mimics to load-bearing constructs.

Significant advancements have been made in engineering polysaccharides for bioprinting applications, yet substantial opportunities remain for further optimisation. The integration of artificial intelligence with experimental biochemistry represents a transformative approach to accelerate the development of polysaccharide-based bioinks. Machine learning algorithms can predict structure–function relationships and optimise modification parameters, while high-throughput experimental validation ensures biological relevance. This synergistic combination promises to enhance the clinical translation potential of polysaccharide biomaterials, facilitating the fabrication of complex, functional tissue constructs with improved physiological accuracy and therapeutic efficacy. Such advancements will be critical for generating viable tissue models and implantable grafts that meet the stringent requirements of regenerative medicine applications.

## Figures and Tables

**Figure 1 jfb-16-00338-f001:**
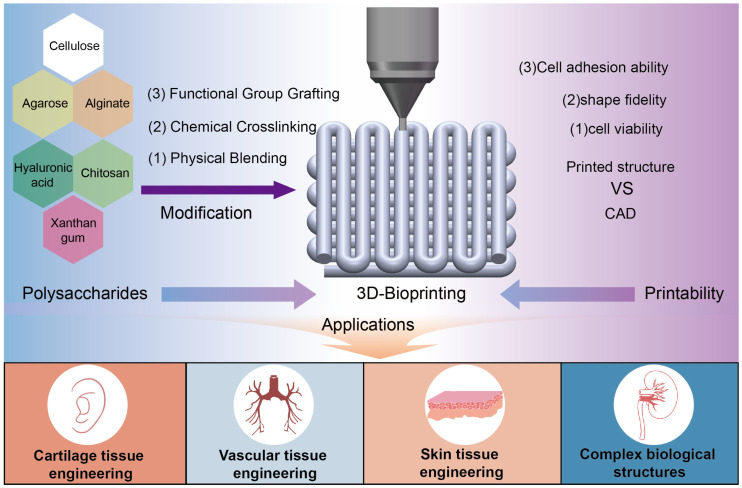
Schematic diagram of polysaccharides modification for 3D bioprinting and tissue engineering applications.

**Figure 2 jfb-16-00338-f002:**
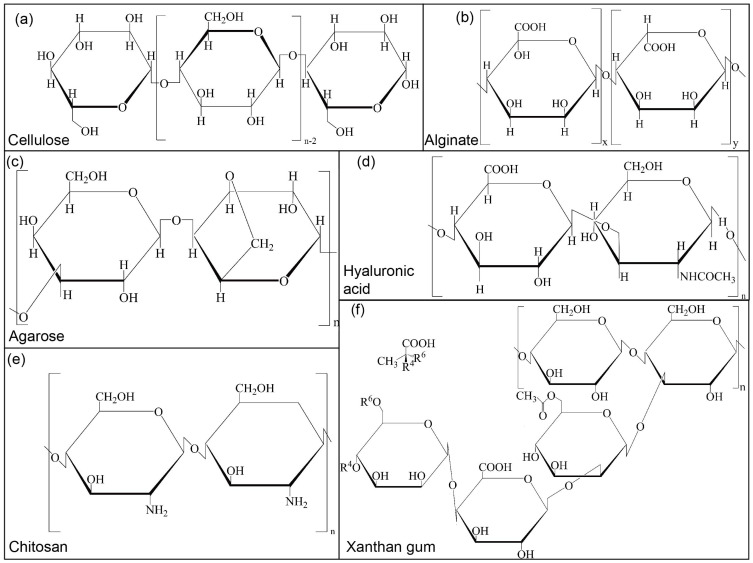
The chemical construction of cellulose (**a**), alginate (**b**), agarose (**c**), hyaluronic acid (**d**), chitosan (**e**), and xanthan gum (**f**).

**Figure 5 jfb-16-00338-f005:**
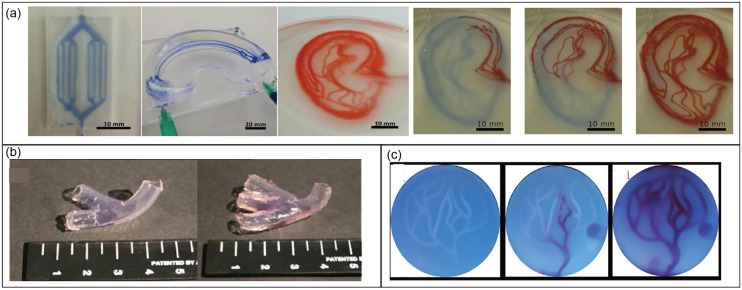
Artificial blood vessel: the fabricated hydrogel microfluidic devices (**a**) (reprinted from Ref. [169]), alginate tubular structure (**b**) (reprinted with permission from Ref. [170], copyright 2022, Elsevier), bioprinted vessel-like networks with nutrient networks (**c**) (reprinted with permission from Ref. [83], copyright 2021, Elsevier).

**Figure 6 jfb-16-00338-f006:**
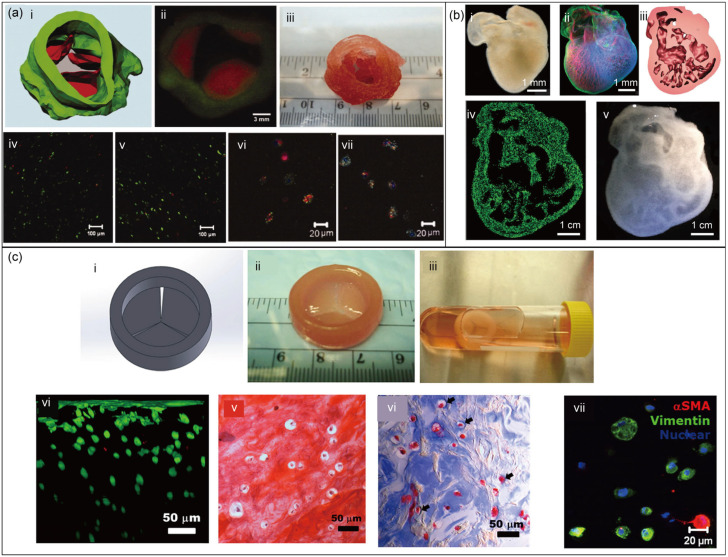
Complex biological structures: (**a**) an aortic valve conduit: (**i**) a micro-CT image, (**ii**) fluorescent image of first printed two layers, (**iii**) a printed aortic valve conduit in colour, (**iv**–**vii**) images of fluorescent staining after 7-day culture (reprinted from Ref. [174]); (**b**) a 3D heart construct of chick embryo: (**i**) a dark-field image of an embryonic chick heart. (**ii**) A 3D image of the chick heart, (**iii**) the 3D CAD model of the embryonic heart, (**iv**) A cross section of the 3D-printed heart in fluorescent alginate (green), (**v**) A dark-field image of the 3D-printed heart (reprinted from Ref. [175]); (**c**) bioprinting of heart valve conduit: (**i**) heart valve model, (**ii**) the printed valve conduit, (**iii**) intact conduit after 7-day static culture, (**iv**–**vii**) staining images (reprinted with permission from Ref. [176], copyright 2014, Elsevier).

**Table 1 jfb-16-00338-t001:** Possible modification methods and 3D bioprinting applications of polysaccharides.

Polysaccharides	Substrate Material	Method or Substance of Modification	Target Substance	Applications	Characteristics	Refs.
Plant polysaccharides	Cellulose	Mechanical shearing	Nanofibrillated cellulose	3D bioprinting and the automated fabrication of complex tissue-mimicking constructs	Accessibility, biocompatibility, and shear-thinning properties	[11]
		Nanometer granulation	Cellulose nanocrystals	Biobased cellulosic scaffold material	Hydrophobicity, oleophilicity, and lipophilicity	[27]
		Alkalization/mercerisation and etherification reactions	Carboxymethyl cellulose	Bioinks for printing bioconstructs	Good viscosity modifier, shear alignment, and shape memory property	[28]
		The open-ring reaction of 1-azido-2,3-epoxypropane (AEP) with hydroxyethyl cellulose	Azido-hydroxy-ethyl cellulose	A novel bioink for bone tissue engineering	Biocompatibility, biodegradability, and printability	[29]
		Hydrophobic modification and hydroxypropyl methylation	Hydroxypropyl methyl cellulose	Bioinks for freeform writing of the millimetric complex tubular structures	Tunable rheological properties, good stability, and compatibility with additives without strong hydrophilic groups	[30]
	Alginate	Tragacanth/hydroxyapatite	Tragacanth/hydroxyapatite modified alginate bioinks	Repair of significant bone tissue defects	Improving compressive strength, viscosity, printability properties, resolution, and shape fidelity	[31]
		Norbornene functionalising	Modular alginate-based bioinks	Construction of complex multi-ink geometries	High cell survivability, stable 3D constructs	[32]
		Oxidised alginate	Degradable alginate-based bioinks	Bioprinting functional cartilage tissue	Rapidly degrade, excellent shape fidelity	[33]
		Molecular weights, concentration, and viscosities	——	Porous bioprinted constructs for bone tissue engineering	Good biocompatibility and tailorable performance	[34]
		ε-polylysine (ε-PL)	ε-polylysine (ε-PL)-modified Alginate-based bioinks (Alg/ε-PL)	Alginate-based scaffolds For the precise and individualised therapy of tissue defects	Excellent self-supporting stability, mechanical stability	[35]
	Agarose	Carboxylated agarose	——	Bioink for Bioprinting of free-standing structures with high Stiffness	Printing high-aspect ratio objects possessing anatomically relevant curvature and architecture	[36]
		2D nanosilicate additives	Nanocomposite agarose	Strong shear-thinning bioinks for extrusion 3D bioprinting applications	Tunable flow properties and bioactivity	[37]
		Polydopamine	Agarose-polydopamine	Hydrogel scaffolds for skin wound healing	Good cell adhesion, biodegradability, and biocompatibility	[38]
Animal polysaccharides	Hyaluronic Acid	Methacrylation of high-molecular-weight hyaluronic acid	Methacrylated hyaluronic acid	Scaffold materials for application in 3D-printed, tissue-engineered bone substitutes	Good primary cell survival and excellent spontaneous osteogenic differentiation in vitro	[39]
		Alginate	A new bioink for cartilage tissue 3D bioprinting	Highly viable and functional bioprinted 3D hybrid structures for Articular cartilage regeneration	Printability, gelling abilities, stiffness, and good degradability	[40]
		Gelatin methacryloyl (GelMA), methacrylated hyaluronic acid (MAHA)	Tunable MAHA-GelMA (metacrylated hyaluronic acid-based hybrid bioinks)	Stereolithographic (SLA) 3D bioprinting	Excellent mechanical strength, printability, and cell-adhesive nature	[41]
		Norbornene functional groups (Nor) and cysteamine hydrochloride (Cys)	Hiol-norbornene photoclick polysaccharide-based bioink	Bioprinting a liver model in vitro	Increased viscoelastic properties, reduced ROS (reactive oxygen species) accumulation, and superior shape fidelity	[42]
	Chitosan	Nanohydroxyapatite (nhap)	Chitosan-nanohap bioinks	3D cellular structures and bone tissue engineering applications	High resolution, shape fidelity, and high printability index	[43]
		Acrylamide (AM), chitosan modified with methacryloyl groups (CHIMA)	CHIMA/AM	A favourable bioink for the DLP-based 3D printing in the field of tissue engineering and regenerative medicine	Enhanced compression strength, improved elasticity, and favourable biocompatibility	[44]
		Grafting chitosan molecular chains with methacryloyl groups	A photocurable chitosan bioink (CHI-MA)	A potential bioink for the DLP and other photocuring-based 3D printing technologies	High resolution, high fidelity, and good biocompatibility	[45]
		Nanostructured bone-like hydroxyapatite(HA)	Chitosan-HA hydrogels	3D bioprinting of tissue constructs	Enabling good mechanical support after printing, providing highly active cell platforms	[46]
		Hyaluronic acid derivatives and Matrigel.	NSC(neural stem cell)-laden scaffold	A neural tissue scaffold	Fast gelation and spontaneous covalent crosslinking capability	[47]
Microbial polysaccharide	Xanthan gum(XG)	Alginate, strontium ions	Crosslinked alginate-xanthan gum blend	Simple cellularized structures and microtissue models to complex organ bioprinting	Noncytotoxic, shear-thinning, and easily sterilizable	[48]
		Calcium-alginate nanoparticles	Alginate-XG hybrid medium	A promising support medium for 3D printing of tissues and organs	Allowing long-term, high resolution, and accurate printing of bio-structures with a high degree of anatomical complexity	[49]
		Carboxymethyl cellulose		Utilising extrusion-based 3D bioprinting	Tunability regarding pore size and mechanical strength optimisation	[50]
		Succinic anhydride	Succinic anhydride (SA)-modified xanthan (XG–SA) derivatives	Promising drug delivery materials for antibacterial applications	Higher storage (G’) and loss (G’) modulus	[51]

## Data Availability

No new data were created or analysed in this study. Data sharing is not applicable to this article.

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
