# Peer review of "Modified Polysaccharides: Potential Biomaterials for Bioprinting"

_jfb, 2025, doi:10.3390/jfb16090338_

Round 1
Reviewer 1 Report
Comments and Suggestions for Authors
Manuscript ID: jfb-3747928
Title: Modified Polysaccharides: Potential Biomaterials for Bioprinting
Journal: Journal of Functional Biomaterials
Authors: Tao Jiang *, Yun Yang, Zening Lin, Yang Hong, Zirong Luo
Overview and general recommendation:
This paper addresses a relevant and current topic, examining the needs that materials used in bioprinting must meet; the advantages and disadvantages of some polysaccharides used in different applications; functionalization options using different methods, such as physical, chemical, and, in some cases, enzymatic methods; and the properties that can be enhanced and improved through these modifications; and the applications and future prospects of modified polysaccharides as potential materials for use in bioprinting.
On the other hand, I would respectfully suggest a more careful review regarding the originality and depth of the key concepts. There are other recent reviews (for example, papers published in Molecular Systems Design & Engineering and Carbohydrate Polymers, between 2023 and 2025).
To strengthen the manuscript's academic contribution and clearly distinguish it from the existing literature, I recommend considering incorporating complementary or less-explored perspectives, such as the critical analysis of unconventional polysaccharides, alternative cross-linking mechanisms, the application of enzymatic modifications, the application of AI, advances in the scale of application (in vitro and in vivo studies), among others.
These improvements will contribute to strengthening the analytical depth of the work, increasing its value as a contribution to the scientific community dedicated to biofabrication and biomaterials from modified polysaccharides.
Below are some suggestions:
- Lines 93-94: It is mentioned, “The use of GenAI for superficial text editing 93 (e.g., grammar, spelling, punctuation, and formatting) does not need to be declared”. It is not related to the content in this paragraph; please check and modify.
- The document is missing citations for Figures 1, .
- Table 1, Refs (40), column “Applications”: This is an extra “for” write in the phrase “A novel bioink for for bone tissue engineering”
- Section 2: include more detailed information about polysaccharides,for example diffferent properties, characterization, among others, as a complement to the information presented.
- Lines 158-160: “based on the number and length variation of G and M (75), the mechanical properties and gelation capabilities of alginates can be modified significantly to meet the requirements of bioprinting.” Can you give more details about this sentence? It could be detailed about what happened (examples) when G is more abundant than M, or vice versa.
- Lines 169-171: “printability was significantly improved through the hybridization of alginate with carboxymethyl cellulose and montmorillonite clay.” Could you provide more details about this report? Specifically, what percentage does this improvement represent, and what is the point of comparison?
- Lines 233-234: “However, researchers have devoted more attention to physical maneuvers.” Can you explain why this is?
- Lines 236-239: “As important parameters for 3D bioprinting, adhesion, mechanical integrity, and degradability can significantly affect the survival and growth of cells. To achieve a favorable printability, it is important to select an appropriate concentration and molecular weight to form hyaluronic hydrogels” Can you give more detailed information about this sentence? For example what are specific values, or a range of values, that are desirable for adhesion, mechanical integrity, degradability for the survival and growth of cells? What is the range of values for the appropriate concentration and molecular weight of HA to form hydrogels?
- Lines 280-282: “Brysch (114) adopted thermomechanical technology to improve the mechanical integrity of chitosan, while Yue combined ultraviolet radiation with ozone treatment to tailor the degradation of chitosan.” Can you provide more details about the improvements obtained in those references? For example, what are modification techniques doing on the chitosan structure that is possible to improve mechanical or degradation properties?
- Lines 283-287: “Chitosan also contains amine, acetylamino, and hydroxyl active sites that are suitable for chemical modification that brings significant improvements in terms of rheological properties, antibacterial properties, and thermal stability (110). The most common chemical modification is acylation with organic acid compounds, which breaks the hydrogen bonds in chitosan, changes its initial crystalline form, and increases its solubility (116).” As the comments above, please provide more details about properties; what kind of organic acids are used to modified chitosan, and how much the solubility is increased.
- Lines 294-299: “Therefore, enzymatic modification has been considered as a safer and more reliable alternative. Aberg (121) used tyrosinase to embed peptides into chitosan, which significantly improved its viscoelasticity. Wang (122) produced novel ternary conjugates of β-lactoglobulin, chitosan, and ferulic acid through laccase induction, resulting in significantly improved antioxidant activity.” Can the reported enzymatic modifications of chitosan replace chemical or physical modifications, or some of them? What other enzymatic modifications have been described for chitosan, and for what purpose or outcome?
- Section 2: What other kinds of enzymatic modifications have been described for the other polysaccharides? Please, complement the information.
- Lines 366-373: “On the other hand, printing parameters including nozzle diameter (150), extrusion pressure (151-154), and translational speed(145, 146) can also greatly affect the quality of printing. The influences of these parameters are always coupled. For example, a higher extrusion pressure typically requires a higher translational speed to achieve a preferable printing quality (155), while over-speeding can cause filament discontinuity and lead to failed printing. Similarly, lowered nozzle diameter would require a higher pressure to initiate the extrusion, together with a lowered speed to compensate the decreased flux in the nozzle.” Can you provide more information details about the values (or a range of values) of the printing parameters? What specific values are considered “over-speeding”? It could be great if you can give more specifications for each polysaccharide.
- Line 387: “Fromm provided a quantitative description” the reference is missing. Please provide the specific reference.
- Page 6, between lines 401 and 402: The “D” equation corresponds to Eq 3.2 or 3.5? Please check and change.
- Lines 403-405: You use the terms “strand printability” and “pore printability”, I suggest to specify the abbreviation in the paragraph as you show it in Eq 3.4 and 3.5.
- Lines 456-458: “Then, based on a weighted evaluation of rheological properties, compression tests, and shape assurance, a suitable formula for biological 3D printing was defined” I suggesto to share the mentioned formula.
- Lines 458-460: “The analysis of cytotoxicity and survival rate indicated that the ink had good biocompatibility and was suitable for cartilage tissue growth.” Could you please give more details about what values or characteristics represent or are considered as good biocompatibility?
- For the references, please review the following:
- Reference 1: The information is incomplete; please complete it.
- Reference 59: Review the reference format.
- Reference 184: The information is incomplete; please complete it.
- Reference 185: The information is incomplete; please complete it.
- Reference 189: The information is incomplete; please complete it.
- Reference 191: The information is incomplete; please complete it.
Thank you for your attention to these suggestions.
Reviewer 2 Report
Comments and Suggestions for Authors
This paper reviews the use of modified polysaccharides as bioinks for 3D printing. The general area has been reviewed dozens of times in the last 5 years, so the focus of this paper needs careful consideration. “Engineering Polysaccharide Biomaterials: Modifications and Crosslinking Strategies for Soft Tissue Bioprinting” (Tembadamani et al, 2025) is the most closely related and does a better job of “systematically summarizes universal modification strategies to address these challenges by tailoring polysaccharides’ physicochemical and biological properties”, which is the stated goal of this review paper. For example, that paper describes the chemical reactions involved in polysaccharide modifications and crosslinking strategies.
This paper lacks those details and universal approaches to modification. Instead, it focuses more on the specific macromolecular structures of various polysaccharides and additives that modify the properties. Chemical or crosslinking schemes are only mentioned in general terms. More detail is needed. What this paper does well is describe quantifiable parameters of bioprinting and applications of bioprinting polysaccharide materials. I recommend that the authors read the many other reviews on the topic and find the niche for this review to make it unique and informative.
In addition, there are some writing issues to correct. While generally well written, there are some odd phrasings. I suggest a native English speaker revise the text. Figure 1 might make more sense if relocated to a section that describes it. And there is a strange statement on GenAI in section 2.
Comments on the Quality of English Languagesome editing recommended
Reviewer 3 Report
Comments and Suggestions for Authors
In this manuscript the authors presented a revie manuscript analysing bioprinting applications of modified polysaccharides. In my opinion the structure of the manuscript has to be improved, both in formatting and content. Here are some suggestions:
- In Fig.1 second row panel on the left, bone tissue engineering is reported but the figures show an ear and a meniscus, thus cartilage tissue engineering seems more appropriate. References of used images are missing in the caption.
- Gelatin and pectin have not been analysed among possible materials.
- In section 3 it is reported: “Since many reviews have detailed the mechanisms of different bioprinting mechanism (142-144), this paper will particularly focus on the qual- ity assessment for extrusion-based bioprinting.” Nevertheless, it is not clear why other bioprinting technologies using polysaccharides have not been analysed, since the cited papers are not related to other technologies for the bioprinting of these materials.
- Paragraphs 3.1 and 3.2 are in my opinion out of the scope of the presented review manuscript.
- Section 4.1 is related to cartilage rather than bone.
Minor comments:
- Pag 2 line 46: ..materials consist of a great..
- Pag 3 line 93-94: I don’t think this is the right position in the manuscript for the generative-AI statement.
- Citation are formatted in different manner throughout the manuscript (e.g., superscript or not). Please always use the same style.
- Check spaces and grammar errors throughout the manuscript.
Round 2
Reviewer 1 Report
Comments and Suggestions for Authors
The revised version of the manuscript has been carefully reviewed. I appreciate the authors’ efforts in addressing the previous comments. The requested corrections have been appropriately incorporated, and the additional modifications made to the manuscript have been clearly explained.
Reviewer 2 Report
Comments and Suggestions for Authors
Concerns have been addressed. The introduction and conclusion sections have been enhanced to clarify the unique focus of this review. The title could be more specific to differentiate this from other reviews but this is not a critical requirement that preclude publishing as is.
Reviewer 3 Report
Comments and Suggestions for Authors
the paper was revised as required and it can be accepted for publication as it